# Anti-Inflammatory and/or Anti-Fibrotic Treatment of MPO-ANCA-Positive Interstitial Lung Disease: A Short Review

**DOI:** 10.3390/jcm11133835

**Published:** 2022-07-01

**Authors:** Hideaki Yamakawa, Yuko Toyoda, Tomohisa Baba, Tomoo Kishaba, Taiki Fukuda, Tamiko Takemura, Kazuyoshi Kuwano

**Affiliations:** 1Department of Respiratory Medicine, Saitama Red Cross Hospital, 1-5 Shintoshin, Chuo-ku, Saitama 330-8553, Japan; 2Department of Respiratory Medicine, Tokyo Jikei University Hospital, 3-25-8 Nishi-shinbashi, Minato-ku, Tokyo 105-8461, Japan; kkuwano@jikei.ac.jp; 3Department of Internal Medicine, Japanese Red Cross Kochi Hospital, 1-4-63-11 Hadaminamimachi, Kochi 780-8562, Japan; yuyutoyo@gmail.com; 4Department of Respiratory Medicine, Kanagawa Cardiovascular and Respiratory Center, 6-16-1 Tomioka-higashi, Kanazawa-ku, Yokohama 236-0051, Japan; baba@kanagawa-junko.jp; 5Department of Respiratory Medicine, Okinawa Chubu Hospital, Okinawa, 81 Miyazato, Uruma 904-2293, Japan; kishabatomoo@gmail.com; 6Department of Radiology, The Jikei University Daisan Hospital, 4-11-1 Izumihoncho Komae-shi, Tokyo 201-8601, Japan; taiki.fukuda@gmail.com; 7Department of Pathology, Kanagawa Cardiovascular and Respiratory Center, 6-16-1 Tomioka-higashi, Kanazawa-ku, Yokohama 236-0051, Japan; tamikobyori@gmail.com

**Keywords:** microscopic polyangiitis, interstitial lung disease, myeloperoxidase antineutrophil cytoplasmic antibody, therapeutic option

## Abstract

The presence of a lung lesion is common in microscopic polyangiitis (MPA), and interstitial lung disease (ILD) can lead to a poor prognosis. Although myeloperoxidase antineutrophil cytoplasmic antibodies (MPO-ANCA) are often present in patients with MPA, patients with ILD and MPO-ANCA positivity but without other manifestations of systemic vasculitis have also been reported. Therefore, the possible association between MPO-ANCA, MPA, and idiopathic ILD remains unclear. This problematic matter has influenced the treatment strategy of MPO-ANCA-positive ILD patients without systemic vasculitis. Clinicians should undertake treatment with careful consideration of the four major causes of death in MPO-ANCA-positive ILD: acute exacerbation of ILD, progressive lung fibrosis, infectious comorbidities, and diffuse alveolar hemorrhage. Further, clinicians need to carefully judge whether inflammation or fibrosis is the dominant condition with reference to the patient’s clinical domain and radiopathological lung features. Recently, anti-fibrotic agents such as nintedanib and pirfenidone were shown to be effective in treating various etiologies associated with ILD and have thus led to the widening of treatment options. In this review, the clinical characteristics, radiopathology, prognosis, and therapeutic options in patients with MPO-ANCA-positive ILD are summarized using limited information from previous studies.

## 1. Introduction

Antineutrophil cytoplasmic antibodies (ANCA) are generally detected in multisystemic diseases, namely ANCA-associated vasculitides such as microscopic polyangiitis (MPA), granulomatosis with polyangiitis (GPA), and eosinophilic granulomatosis with polyangiitis (EGPA) [1,2,3]. The presence of a lung lesion is a common and important clinical feature in ANCA-associated vasculitides [2,4]. However, interstitial lung disease (ILD) is rarely seen in patients with GPA and EGPA [1,2,3,4], but the presence of ILD is a common and important clinical feature in MPA [1,3,4]. In addition, MPO-ANCA have been frequently identified in patients with ILD without multisystemic disease. A recent report showed that ILD was diagnosed before (52%) or simultaneously (39%) with ANCA-associated vasculitides [5]. Therefore, its association with ILD has often been discussed [6,7,8] because ILD can lead to a poor prognosis in patients with MPA [5,9,10,11]. In addition, patients with pulmonary fibrosis and ANCA positivity but without other manifestations of systemic vasculitis have also been reported, which was sometimes diagnosed as “pulmonary-limited type of ANCA-associated vasculitis” [8,12]. Furthermore, ANCA-positive conversion has been described in patients with an initial diagnosis of idiopathic ILD, with manifestations of systemic vasculitis occurring in some patients [13,14]. However, it is unclear whether ILD observed in the presence of only MPO-ANCA differs from idiopathic ILD and ILD with overt MPA [14]. Needless to say, this problematic matter has influenced the treatment strategy of MPO-ANCA-positive ILD patients without systemic vasculitis as a form of idiopathic ILD. Therefore, the aim of the present review was to assess the characteristics of MPO-ANCA-positive ILD using limited information from previous studies in this field and then focus on the future treatment of MPO-ANCA-positive ILD.

## 2. Prevalence and Clinical Manifestations

The prevalence of MPO-ANCA positivity ranges from 1.7% to 22.2% in patients with idiopathic ILD [8,13,15,16]. From the viewpoint of MPA, the prevalence of ILD ranges from 2.7% to 47.4% in patients with MPA [8,17,18,19,20,21,22,23,24,25]. ILD is more frequently associated with MPO-ANCA positivity in Japanese patients than in Western patients [5,26]. MPO-ANCA-positive ILD is usually observed in patients older than 65 years old as idiopathic pulmonary fibrosis (IPF) [17].

Conversion to MPO-ANCA positivity occurs in patients initially diagnosed as having idiopathic ILD at a reported rate of prevalence of 3.3% to 5.7% [13,15]. Moreover, a quarter of the patients with MPO-ANCA positivity at the diagnosis of idiopathic ILD or with conversion to MPO-ANCA positivity during follow-up developed MPA [8,13,15]. To put it differently, clinicians should be careful to note that MPA develops at a certain incidence rate in MPO-ANCA-positive ILD patients [8,10,13,15].

## 3. Morphological Domain

### 3.1. Radiological Findings

In previous reports, radiological analysis, especially by high-resolution computed tomography (HRCT), was used for the classification of patterns [27,28] in patients with MPO-ANCA-positive ILD. In patients with MPA, the most frequently occurring pattern was usual interstitial pneumonia (UIP) (50–78%), followed by nonspecific interstitial pneumonia (NSIP) (7–58%) and others (13–31%) [14,29,30,31,32]. MPO-ANCA-positive ILD without systemic vasculitis (i.e., idiopathic ILD) also showed a similar tendency (UIP: 12.9–53.9%, non-UIP: 13.6–58.1%) [31,32]. Honeycombing was seen in about 30% of the MPO-ANCA-positive ILD patients regardless of whether they had MPA-ILD or idiopathic ILD [30,31,32].

### 3.2. Pathological Findings

In studies that specifically described the characteristics of patients with ILD associated with MPA and MPO-ANCA-positive idiopathic ILD, UIP was the most frequent pathological pattern (46–100%), and marked dense fibrosis and areas of honeycombing were common in these specimens [17,20,21,33,34,35]. In addition, a high incidence of findings such as lymphoid hyperplasia, organizing pneumonia, pleuritis, and bronchiolitis is a characteristic observed in ILD associated with MPA and MPO-ANCA-positive idiopathic ILD [18,33,34].

### 3.3. Differences between MPO-ANCA-Positive ILD and UIP/IPF

For the reason stated above in regard to radiopathological findings, similar to UIP/IPF, MPO-ANCA-positive ILD most commonly showed a UIP pattern and honeycombing as rheumatoid arthritis (RA)-ILD [9,17,36], whereas ILD in patients with systemic sclerosis and myositis is predominantly associated with a non-UIP pattern [9,17,37]. A recent study reported that the single nucleotide polymorphism (SNP) rs35705950T was more strongly associated with UIP than with non-UIP [38]. The SNP rs35705950 (G/T) in the promotor region of MUC5B, which encodes mucin 5B, is a strong genetic factor for UIP/IPF and RA-UIP [39,40]. The MUC5B promoter variant rs35705950 might also be a strong risk factor for MPA-ILD [41]. Therefore, these findings suggest at least a partially shared genetic susceptibility between MPO-ANCA-positive ILD and UIP/IPF.

However, some previous studies reported that MPO-ANCA-positive ILD (including MPA-ILD) and UIP/IPF showed different radiopathological features, despite having the same basic UIP pattern [30,31,33,34,35]. Radiologically, increased attenuation around areas of honeycombing and traction bronchiectasis (Figure 1A) was more frequently seen in MPO-ANCA-positive ILD than in UIP/IPF [30,31,34]. We reported that an anterior upper lobe honeycomb-like lesion, which represents a concentration of cystic air spaces within the anterior aspect of the upper lobes (Figure 1B), might be found with higher frequency in MPO-ANCA-positive ILD and RA-ILD than in other etiologies associated with ILD [31,42]. Pathologically, MPO-ANCA-positive ILD even with a UIP pattern showed more prominent inflammatory cell infiltration and cellular bronchiolitis, unlike UIP/IPF (Figure 2A–E) [8,14,17,33,34,35]. In other words, these pathological features correspond well to the presence of a radiological feature such as increased attenuation around areas of honeycombing and traction bronchiectasis [30,31,34]. Therefore, these features may be associated with a better response to anti-inflammatory therapies in MPO-ANCA-positive ILD patients, and clinicians should be careful to note the development of MPA in patients having these features [31,34] (Table 1).

## 4. Prognosis and Causes of Death

Studies of previous cohorts of patients with MPA-ILD reported median survival times of 4.1–11 years and 5-year survival rates of 46–60%, although long-term outcomes are less well reported [10,31,32,34,43]. In MPO-ANCA-positive ILD without systemic vasculitis as a form of idiopathic ILD, whether there is a difference in survival time and prognosis between patients with MPA-ILD and those with idiopathic ILD is currently controversial [31,32]. As a caveat, there is a high probability that patients with MPO-ANCA-positive ILD even if showing a UIP pattern that had longer survival than those with UIP/IPF [10,13,34]. The poor prognostic factors reported included older age, higher ANCA titers, higher levels of inflammatory markers (i.e., erythrocyte sedimentation rate [ESR] and C-reactive protein [CRP]), lower forced vital capacity, and anterior upper lobe honeycomb-like lesions [10,13,14,16,31,32,34,44].

The major causes of death in patients with MPO-ANCA-positive ILD were acute exacerbation (AE) of ILD (10–25.5%), chronic progressive fibrosing ILD (PF-ILD) (7.8–19%), infectious disease (19.2–20%), and diffuse alveolar hemorrhage (DAH) (4.3–30%) [10,11,31]. As a cautionary point, it is possible that patients with MPO-ANCA-positive idiopathic ILD may present with DAH as the initial manifestation of MPA development, which is a serious condition, as DAH can be life threatening [10,13,15,31]. Taken together, these findings suggest that clinicians should create a decision-making strategy for the treatment of MPO-ANCA-positive ILD with an awareness of the presence of these four major causes of death (i.e., AE of ILD, PF-ILD, infectious disease, and DAH).

## 5. Therapeutic Assessment of MPO-ANCA-Positive ILD

### 5.1. ILD Patients with MPA

When ILD associated with MPA is diagnosed, treatment should be based on organ involvements of vasculitis [8]. Treatment of patients with MPA should include induction therapy using highly potent anti-inflammatory agents (e.g., corticosteroid, cyclophosphamide, rituximab, mycophenolate, and azathioprine) [9,14,17,32,44]. However, no studies have shown whether therapy to treat MPA itself is also effective on ILD. Unfortunately, ILD often independently leads to progressive lung fibrosis [45].

### 5.2. MPO-ANCA-Positive ILD Patients without Systemic Vasculitis as Idiopathic ILD

Initially, when MPO-ANCA-positive ILD without systemic vasculitis is considered, clinicians should determine whether the patient has a UIP pattern on HRCT. In patients with NSIP or an organizing pneumonia pattern, anti-inflammatory agents (mainly as corticosteroids) should be started to treat these conditions as idiopathic ILD [14,17].

Second, the therapeutic assessment of MPO-ANCA-positive ILD with a UIP pattern is most difficult and controversial due to the lack of evidence [14]. This also includes progressive pulmonary fibrosis (i.e., PF-ILD) with MPA despite anti-inflammatory therapies directed at treating MPA itself. Two main factors are important when considering the decision to start or switch treatment, whether inflammatory change or fibrotic change in ILD is the dominant factor [46].

One factor favoring inflammatory change is the presence of MPA-like disease (mock MPA), as indicated by recurrent unexplained fever and/or higher CRP/ESR and an upward trend in the MPO-ANCA titer. Moreover, the second factor is increased attenuation around fibrotic change seen on HRCT and prominent inflammatory cell infiltration in pathological lung lesions [1,2,3,47,48]. Regarding the former (i.e., mock MPA), if patients with MPO-ANCA-positive idiopathic ILD develop MPA, they should be treated with anti-inflammatory therapy because untreated MPA (particularly that with DAH) is normally progressive and fatal [13,31]. In terms of the latter as a radiopathological finding, anti-inflammatory therapy can be an effective course of treatment even if a UIP pattern is present, as shown in Figure 3 and Figure 4. Therefore, if these conditions are present, anti-inflammatory therapy should be considered.

In contrast, the factors favoring fibrotic change are nonincreasing levels of CRP or ESR, a decrease or leveling off of the MPO-ANCA titer, expansion of fibrotic change as reticulation with traction bronchiectasis and honeycombing on HRCT without increasing attenuation (i.e., ground-glass opacity [GGO] and consolidation), and pathologically patchy fibrosis as a UIP pattern and/or fibroblastic foci in a lung specimen without prominent inflammatory cells [1,2,3,30,31,33,34,47,48,49,50]. As a caveat, GGO, which is a finding typically associated with inflammation or infiltration, does not always represent reversible lung disease, but rather in some cases, it may represent microscopic fibrosis [51]. Therefore, lung biopsy may be useful to assess whether inflammatory or fibrotic change is dominant [36,49,52]. Importantly, recent reports have shown that all etiologies associated with ILD can lead to a poor prognosis, particularly in cases in which a component of UIP is found to a certain degree [53,54,55]. In other words, some cases represent the natural evolution of the inflammatory change of ILD into fibrotic change [56]. Adegunsoye et al. noted that honeycombing represents a progressive fibrotic ILD phenotype regardless of underlying conditions such as connective tissue disease, idiopathic ILD, or fibrotic hypersensitive pneumonia [57]. In addition, AE of ILD can affect all patients with ILD but apparently occurs more frequently in patients with an underlying UIP pattern, which is a most harmful event and can be a major cause of death [58,59]. Therefore, the presence and/or expansion of fibrotic change particularly in the UIP pattern appearing as reticulation with traction bronchiectasis and honeycombing during follow-up has an important prognostic implication in all etiologies associated with ILD [36,40,53]. If we determine that fibrotic change is dominant, we consider starting an anti-fibrotic therapy such as nintedanib or pirfenidone, which has been approved for the treatment of IPF [60,61,62]. The INBUILD study showed the effectiveness of nintedanib on PF-ILD other than IPF, and nintedanib might have the potential to suppress the AE of ILD [63]. Post hoc analysis of the INBUILD study suggested a treatment benefit of nintedanib across all subgroups of patients with PF-ILD, including autoimmune ILD, although the trial was not powered to provide evidence to address this specific question [64]. In the RELIEF study, although the quality of evidence was rated as low, pirfenidone also significantly suppressed the deterioration of forced vital capacity in patients with progressive fibrotic ILD due to four diagnoses: connective tissue disease-associated ILDs, fibrotic NSIP, fibrotic hypersensitivity pneumonitis, and asbestos-induced lung fibrosis [65]. Taken together, expanding fibrotic change in patients with a UIP pattern can lead to the high risk of PF-ILD and AE of ILD, which means a poorer prognosis even in those with MPO-ANCA-positive ILD. Therefore, starting anti-fibrotic therapy such as nintedanib should be considered in such a case (Figure 5, same patient as in Figure 2).

Additionally, considering therapeutic assessment from a different perspective, anti-inflammatory therapy could be harmful in some cases. The importance of separating IPF from ILD associated with specific systemic conditions such as those related to connective tissue diseases has been emphasized [33,66]. It is recommended that IPF patients should not be treated with anti-inflammatory therapies [66,67]. A recent report also showed anti-inflammatory therapy as indicated by steroid use to have a higher risk of developing PF-ILD in connective tissue disease-associated ILD [68]. In RA-ILD, prednisolone doses higher than 10 mg are associated with a higher risk of infectious comorbidity [69]. As also with MPA-ILD in elderly patients, the administration of high-dose corticosteroid therapy is a significant risk factor for severe infection, which is a main cause of death [70]. In general, corticosteroids had been considered to result in a higher risk of chronic pulmonary infection, such as nontuberculous mycobacteria and pulmonary aspergillosis [71,72]. Because the presence of emphysema may also be a risk factor for chronic pulmonary infection in ILD patients [73,74], when considering the initiation or strength of anti-inflammatory therapy, clinicians should pay particular attention to those patients having pulmonary emphysema with ILD to some extent. Moreover, Kawamura et al. reported that corticosteroid use was associated with an increased risk of AE of ILD even in a dose-dependent manner at low doses [75]. Although we cannot say for certain, incomplete anti-inflammatory therapy may cause harmful events such as infectious disease and AE of ILD that can lead to a poor prognosis. Taken together, when it is difficult to select between anti-inflammatory and anti-fibrotic therapy, anti-fibrotic therapy should be given priority in the patients at high risk for infection and AE of ILD such as initiation or long-term administration of moderate- to high-dose corticosteroid, concomitant to a certain extent with pulmonary emphysema (Figure 6).

A prospective study should clarify the natural history of MPA-ILD and the role of ANCA positivity in patients with idiopathic ILD and offer a better therapeutic approach to treating the different groups of patients with this condition [76].

## 6. Conclusions

MPO-ANCA-positive ILD is not uncommon in any case of MPA-ILD or idiopathic ILD. Because some patients with MPO-ANCA positivity at the diagnosis of idiopathic ILD or with MPO-ANCA-positive conversion during follow-up develop MPA, clinicians should pay particular attention to the presence of DAH because this complication can be fatal. In patients with MPA-ILD, MPA itself should first be stabilized with anti-inflammatory therapies. When MPO-ANCA-positive ILD with UIP pattern including PF-ILD with MPA is present despite anti-inflammatory therapies, clinicians need to carefully judge whether inflammation due to fibrosis is the dominant condition. When it is difficult to select between anti-inflammatory and anti-fibrotic therapy, anti-fibrotic therapy should be given priority in the patients at high risk for infection and AE of ILD. Because MPO-ANCA-positive ILD can result in the development of systemic vasculitides and/or a poor prognosis, the monitoring of patients over time is important. Multidisciplinary discussion among the pulmonologist, rheumatologist, radiologist, and pathologist can be useful to correctly approach the therapeutic assessment of patients with MPO-ANCA positivity.

## Figures and Tables

**Figure 1 jcm-11-03835-f001:**
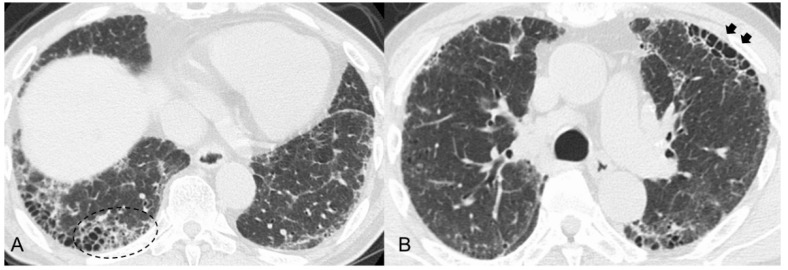
(**A**) High-resolution computed tomography (HRCT) scan shows increased attenuation around honeycombing and traction bronchiectasis (dashed circle). (**B**) HRCT shows anterior upper lobe honeycomb-like lesion as a concentration of cystic air spaces within the anterior aspect of the upper lobes (black arrows).

**Figure 2 jcm-11-03835-f002:**
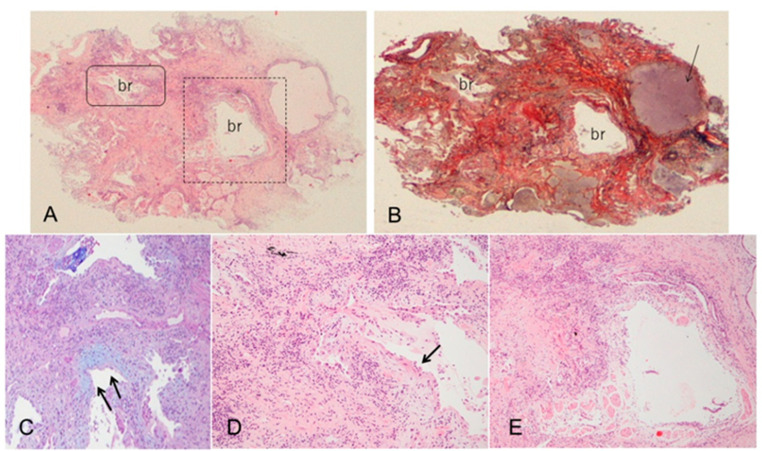
Transbronchial lung cryobiopsy in a case of MPO-ANCA-positive ILD. (**A**) Hematoxylin and eosin staining shows that the lesion is characterized by peribronchial dense fibrosis of alveoli with architectural destruction as a framework of usual interstitial pneumonia (UIP) pattern along with inflammatory cells in the interstitial tissue (×1) (Br, bronchus). (**B**) Elastica van Gieson staining shows partial loss of elastic lamina of a bronchiole as bronchiolitis and cystic air spaces filled with mucus (arrow) (×1). (**C**,**D**) A high-power magnification view of the closed square in (**A**) shows dilated bronchioles with subepithelial fibroblastic foci (arrows) and moderate infiltration of lymphocytes and plasma cells in the fibrosis ((**C**): ×2.5; Alcian blue staining, (**D**): ×2.5; hematoxylin and eosin staining). (**E**) A magnified view of the dashed square in (**A**) shows a dilated bronchiole revealing shedding of epithelial cells, subepithelial fibrosis, and infiltration of lymphocytes and plasma cells, which defines bronchiolitis (×2.5; hematoxylin and eosin staining).

**Figure 3 jcm-11-03835-f003:**
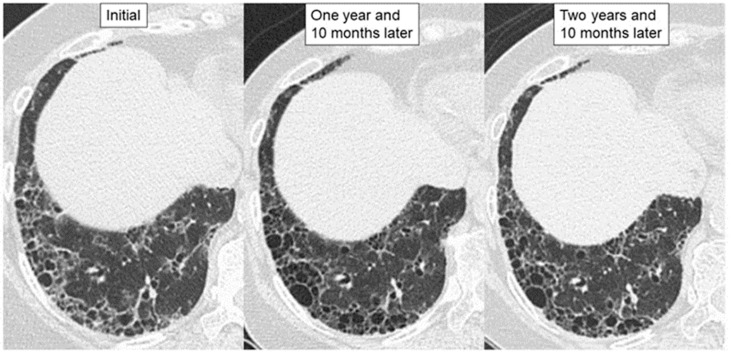
High-resolution computed tomographic images of a patient with a stable course of MPA-ILD with UIP pattern for less than 3 years. The patient received anti-inflammatory therapies, after which wall thickening in the areas of honeycombing had improved, and the fibrotic lesion did not extend without requiring administration of an anti-fibrotic agent.

**Figure 4 jcm-11-03835-f004:**
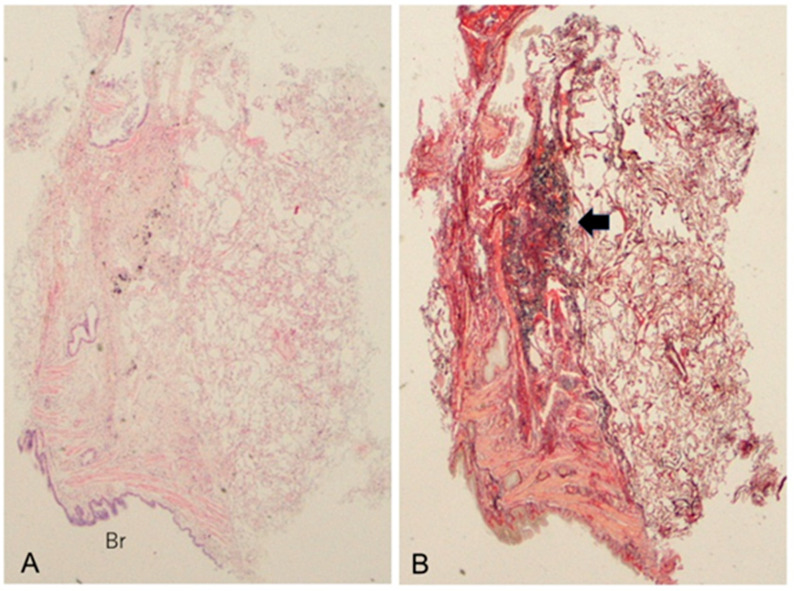
Pathological findings of the same patient with MPA-ILD shown in Figure 3. The transbronchial lung cryobiopsy specimens revealed the presence of dense fibrosis with elastosis apposed to a bronchus, located at the perilobular area (arrow), and an area of normal alveoli can be also seen, which makes up the framework of the UIP pattern ((**A**): ×1; hematoxylin and eosin staining; Br, bronchus. (**B**): ×1; Elastica van Gieson staining).

**Figure 5 jcm-11-03835-f005:**
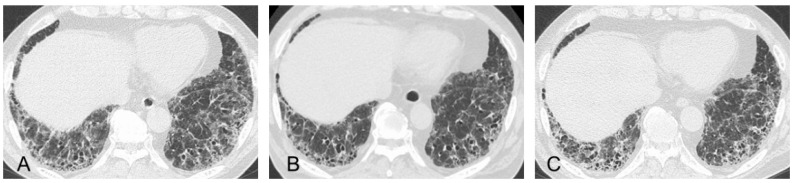
Radiologic course of the same patient with MPO-ANCA-positive ILD shown in Figure 2. (**A**) At the time of transbronchial lung cryobiopsy, high-resolution computed tomographic (HRCT) imaging showed subpleural reticulations with traction bronchiectasis as the framework of UIP and ground-glass opacities (GGOs) to some extent in the bilateral lung field. (**B**) At six months after anti-inflammatory treatments, the findings regarding GGOs were improved. (**C**) At six more months later, HRCT showed a moderate increase in disease extent of the reticulations and traction bronchiectasis as indications of progressive fibrosis, for which therapy with the anti-fibrotic agent nintedanib was initiated.

**Figure 6 jcm-11-03835-f006:**
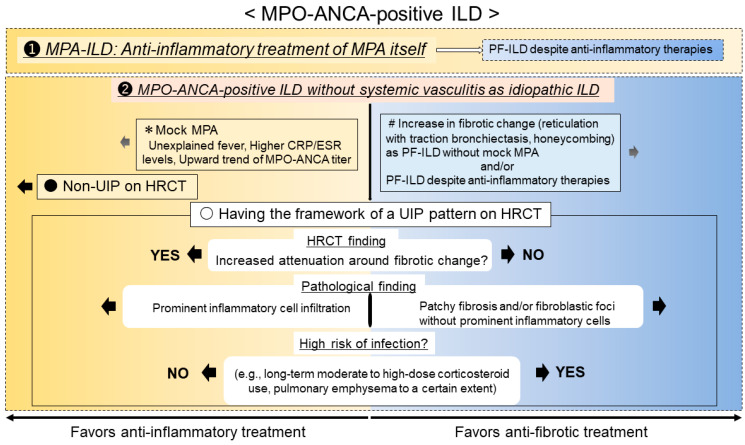
Proposed protocol for the treatment of MPO-ANCA-positive ILD. MPA, microscopic polyangiitis; MPO-ANCA, myeloperoxidase antineutrophil cytoplasmic antibody; ILD, interstitial lung disease; CRP, C-reactive protein; ESR, erythrocyte sedimentation rate; PF-ILD, progressive fibrosing ILD; HRCT, high-resolution computed tomography; UIP, usual interstitial pneumonia.

**Table 1 jcm-11-03835-t001:** Characteristics and HRCT/pathological findings in MPO-ANCA-positive ILD and IPF.

	MPO-ANCA-Positive ILD/MPA-ILD	IPF
**Age**	Commonly older than 65 years old (patients younger than 50 years old are rare)	
**Smoking**	The majority of patients have a history of past cigarette smoking	
**Sex**	Male (45.2–66.7%) ≒ Female (33.3–54.7%)	Male (>70%) > Female
**HRCT findings**	Framework: Subpleural and basal predominant distribution is often heterogeneous	
	Increased attenuation around honeycombing and traction bronchiectasis (19–39%)	―
**Pathological findings**	Framework: Dense fibrosis with architectural distortion, predominant subpleural and/or paraseptal distribution of fibrosis	
	More prominent inflammatory cell infiltration and cellular bronchiolitis compared with IPF	―

## Data Availability

Not applicable.

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
