# Peer review of "Anti-Inflammatory and/or Anti-Fibrotic Treatment of MPO-ANCA-Positive Interstitial Lung Disease: A Short Review"

_jcm, 2022, doi:10.3390/jcm11133835_

Round 1

Reviewer 1 Report

I had the pleasure of reading the review by Yamakawa H. et al. This is an interesting paper that summarizes the characteristics of ILD positive MPO-ANCA. This topic is of great interest and topical, I have only a few minor revisions to suggest.

INTRODUCTION

-Lines 40-43: It should be kept in mind that the subject of the paper is MPO-ILD (as it is rightly specified in the title). In my opinion, the introduction should bigin by describing MPO and MPO-ILD; then describing ILD related to ANCA associated vasculitis such as MPA but also mentioning GPA (the presence of MPO and ILD is well described in litterature). Currently, the introduction rises some confusion regarding the subject of the review.

-Lines 48-52: you should cite an important clinical behaviour of the ANCA associate ILD: ILD may represent the first and the only clinical manifestation of AAV in its beginning (doi: 10.1097/MD.0000000000000217; DOI: 10.1016/j.jaut.2019.102338).

PREVALENCE AND CLINICAL MANIFESTATION

-Lines 58-60: the subject of the review is not clear. Please, first describe the features of MPO-ILD, then those of MPA-ILD and, at the end, the comment on the MPO conversion in IPF patients (ok lines 64-69).

MORPHOLOGICAL DOMAIN

-Lines 72-74: you should cite the classification of ILD pattern (radiological and phatological) in rheumatologic diseases, that is the same of IPF (https://doi.org/10.1016/S2213-2600(17)30433-2.)

-Lines 91-93: ...patients with SSc and myositis AND OTHER CONNECTIVE TISSUE DISEASES is predominantly... 

-Chapter 3.3: this chapter is of great interest to clinicians. Please, you consider making a table that summarizes the main differences (radiological, pathological, clinical, genetic, etc.) between IPF, MPO-ILD, MPA-ILD. Thank you.

THERAPEUTIC ASSESSMENT

-Lines 155-159: unfortunally, no studies demonstrated that vasculitis therapy is also effective on the interstitial pneumonia. In established MPA-ILD, the systemic vasculitis must be treated to induce disease remission, but we do not know if this is sufficient to treat the lung as well. Unfortunately, lung disease often has an independent course. what you describe is the current clinical practice, without strong evidences. Please, reword chapter 5.1.

-Lines 215-217: Were there vasculitis in the subpopulations of the INBUILDstudy? Please, investigate this topic.

CONCLUSIONS

-Line 270: "MPA itself should first be stabilized". YES!!! this is the goal. DMARDs are effective in MPA, but maybe not in ILD. Unfortunately, we still have little scientific evidence for the treatment of AAV-ILD or MPO-ILD. 

-Please, highlight the importance of follow-up in patients with isolated ILD or MPO-ILD, as they can develop systemic vasculitis with poor prognosis.

REFERENCES

-Please, consider adding in the references: doi: 10.3390/jcm10122548. PMID: 32324122 

Thank you for your nice review.

Author Response

Title: Anti-inflammatory and/or Anti-fibrotic Treatment of MPO-ANCA-positive Interstitial Lung Disease: A Short Review

We appreciate your giving us the opportunity to revise our manuscript. We are grateful to you and the reviewers for your helpful comments to revise our manuscript. We have made every effort to prepare this updated version of our paper accordingly.

Reviewer 1

I had the pleasure of reading the review by Yamakawa H. et al. This is an interesting paper that summarizes the characteristics of ILD positive MPO-ANCA. This topic is of great interest and topical, I have only a few minor revisions to suggest.

âž¡Reply: Thank you for your comments. We have made every effort to prepare the revised version of our paper in accordance with your comments. The revised text is written in red color in the manuscript.

INTRODUCTION

  • -Lines 40-43: It should be kept in mind that the subject of the paper is MPO-ILD (as it is rightly specified in the title). In my opinion, the introduction should bigin by describing MPO and MPO-ILD; then describing ILD related to ANCA associated vasculitis such as MPA but also mentioning GPA (the presence of MPO and ILD is well described in litterature). Currently, the introduction rises some confusion regarding the subject of the review.

âž¡Reply: Thank you for your comments. As suggested, we revised the beginning of the introduction to improve the understanding of the readers.

  • -Lines 48-52: you should cite an important clinical behaviour of the ANCA associate ILD: ILD may represent the first and the only clinical manifestation of AAV in its beginning (doi: 10.1097/MD.0000000000000217; DOI: 10.1016/j.jaut.2019.102338).

âž¡Reply: Thank you for your suggestion. We added this information in this manuscript as follows:

“A recent report showed that ILD was diagnosed before (52%) or simultaneously (39%) with ANCA-associated vasculitides [5].”

  • PREVALENCE AND CLINICAL MANIFESTATION

-Lines 58-60: the subject of the review is not clear. Please, first describe the features of MPO-ILD, then those of MPA-ILD and, at the end, the comment on the MPO conversion in IPF patients (ok lines 64-69).

âž¡Reply: Thank you for your suggestion. We revised this as you suggested.

  • MORPHOLOGICAL DOMAIN

-Lines 72-74: you should cite the classification of ILD pattern (radiological and phatological) in rheumatologic diseases, that is the same of IPF (https://doi.org/10.1016/S2213-2600(17)30433-2.)

âž¡Reply: Thank you for your comments. We added references to the HRCT classification.

  • -Lines 91-93: ...patients with SSc and myositis AND OTHER CONNECTIVE TISSUE DISEASES is predominantly... 

-Chapter 3.3: this chapter is of great interest to clinicians. Please, you consider making a table that summarizes the main differences (radiological, pathological, clinical, genetic, etc.) between IPF, MPO-ILD, MPA-ILD. Thank you.

âž¡Reply: Thank you for your comments. We added the Table as this matter.

  • THERAPEUTIC ASSESSMENT

-Lines 155-159: unfortunally, no studies demonstrated that vasculitis therapy is also effective on the interstitial pneumonia. In established MPA-ILD, the systemic vasculitis must be treated to induce disease remission, but we do not know if this is sufficient to treat the lung as well. Unfortunately, lung disease often has an independent course. what you describe is the current clinical practice, without strong evidences. Please, reword chapter 5.1.

âž¡Reply: Thank you for your important suggestion. This is exactly as you described. Therefore, we added the following: “However, no studies have shown whether therapy to treat MPA itself is also effective on the ILD. Unfortunately, ILD often independently leads to progressive lung fibrosis [46]”.

  • -Lines 215-217: Were there vasculitis in the subpopulations of the INBUILDstudy? Please, investigate this topic.

âž¡Reply: Thank you for your question. We do not have exact information on this matter so we added the following sentence.

“Post hoc analysis of the INBUILD study suggested a treatment benefit of nintedanib across all subgroups of the patients with PF-ILD, including autoimmune ILD, although the trial was not powered to provide evidence to address this specific question [65].”

  • CONCLUSIONS

-Line 270: "MPA itself should first be stabilized". YES!!! this is the goal. DMARDs are effective in MPA, but maybe not in ILD. Unfortunately, we still have little scientific evidence for the treatment of AAV-ILD or MPO-ILD. 

-Please, highlight the importance of follow-up in patients with isolated ILD or MPO-ILD, as they can develop systemic vasculitis with poor prognosis.

âž¡Reply: Thank you for your comments. We added this point in the conclusion paragraph as follows:

“Because MPO-ANCA-positive ILD can result in the development of systemic vasculitides and/or a poor prognosis, the monitoring of patients over time is very important.”

  • REFERENCES

-Please, consider adding in the references: doi: 10.3390/jcm10122548. PMID: 32324122 

âž¡Reply: Thank you for your suggestion. We added this reference.

Reviewer 2 Report

In this review article, the authors Hideaki Yamakawa et. al., have raised a question for the current treatment strategy for MPO-ANCA-positive ILD patients and and proposed an alternate approach that can be used for treating MPO-ANCA-positive ILD patients, differently than MPA-ILD patients. The authors mentions the need to do so, because there is an uncertainty in differentiating ILD caused by MPO-ANCA-positivity versus ILD caused by MPO. According to authors, this uncertainty raises the question about the treatment strategy used to treat MPO-ANCA-positive patients.

A] Strength:

1) The review article provides a good understanding for a reasonably broader range of readers.

2) The review is well-written and delivers the message appropriately and clearly by providing information on prevalence, previous findings and reports, a comparison among previous reports, and based on them a couple of suggestions.

B] Minor suggestion

1) The authors have cited relevant literature for the information from previous case studies and reports on MPA-ILD and MPO-ANCA-positive ILD. If available, addition of information from relevant previous clinical trial outputs for treating MPA-ANCA-positive ILD patients with current treatment strategy and its pitfalls may help to strengthen the review article.

Author Response

Reviewer ï¼’

In this review article, the authors Hideaki Yamakawa et. al., have raised a question for the current treatment strategy for MPO-ANCA-positive ILD patients and and proposed an alternate approach that can be used for treating MPO-ANCA-positive ILD patients, differently than MPA-ILD patients. The authors mentions the need to do so, because there is an uncertainty in differentiating ILD caused by MPO-ANCA-positivity versus ILD caused by MPO. According to authors, this uncertainty raises the question about the treatment strategy used to treat MPO-ANCA-positive patients.

A] Strength:

1) The review article provides a good understanding for a reasonably broader range of readers.

2) The review is well-written and delivers the message appropriately and clearly by providing information on prevalence, previous findings and reports, a comparison among previous reports, and based on them a couple of suggestions.

âž¡Reply: Thank you for your comments. We have made every effort to prepare the revised version of our paper in accord with your suggestion. The revised text is written in red color in the manuscript.

B] Minor suggestion

1) The authors have cited relevant literature for the information from previous case studies and reports on MPA-ILD and MPO-ANCA-positive ILD. If available, addition of information from relevant previous clinical trial outputs for treating MPA-ANCA-positive ILD patients with current treatment strategy and its pitfalls may help to strengthen the review article.

âž¡Reply: Thank you for your suggestion. As you mentioned, our review article summarized limited information from previous studies, but the treatment of MPO-ANCA-positive ILD remains an unmet clinical need. Therefore, we added the following sentence referring to a previous review:

“A prospective study should clarify the natural history of MPA-ILD and the role of ANCA positivity in patients with idiopathic ILD and offer a better therapeutic approach to treating the different groups of patients with this condition [77]”.
